# The sensitivity of TANDEM – A new measure of trauma competence

**Rolf Gjestad[1,2], Dag Ø. Nordanger[3,4]\*, Alina Coman[3], Anca Maria Yttri[3]**

1 Centre for Crisis Psychology, Faculty of Psychology, University of Bergen, Bergen, Norway, 2 Research Department, Division of Psychiatry, Haukeland University Hospital, Bergen, Norway, 3 Faculty of Health Sciences, Department for Nursing Sciences and Health Promotion, Unit for Mental Health, Substance Use and Addiction, Oslo Metropolitan University, Oslo, Norway, 4 RVTS West, Haukeland University Hospital, Bergen, Norway

\* dagno@oslomet.no

## Abstract

### Background

Despite international standards outlining the competence areas to be included in educational programs on trauma-informed care, a Norwegian expert group identified a lack of tools to measure the learning outcomes of such programs adapted to the Norwegian context. In response, they developed the Trauma and Development Education Monitor (TANDEM). This study examines the sensitivity of the instrument.

### Methods

The study is based on pre- and post-test survey responses to TANDEM's 55 items, covering the domains of Readiness, Agency, Reflexivity, Knowledge, Skills, and Work Culture, collected from 370 students across four cohorts enrolled in a three-week trauma course at Oslo Metropolitan University. Confirmatory factor analysis (CFA) with measurement invariance analyses were conducted to evaluate configural, weak, strong, and strict invariance in the reflective dimensions Readiness, Agency, and Reflexivity. Latent growth curve (LGC) models were then applied to analyze changes in outcome scores from pre- to post-program. Finally, LGC levels and changes in outcomes were predicted by participant background variables, i.e., workplace type, target age group, role type, work seniority, and prior trauma training.

### Findings

TANDEM demonstrated sensitivity in detecting significant increases in scores from before to after the course, both for the instrument as a whole and across all domains, most notably for Knowledge and Skills. Some measurement invariance issues were identified in the Agency and Reflexivity domains, indicating interpretation problems. Significant pre–post changes for the total instrument were observed regardless

**Data availability statement:** The full data matrix is available at https://doi.org/10.18712/NSD-NSD3333-V1 in the Sikt repository (Sikt – Norwegian Agency for Shared Services in Education and Research).

**Funding:** The author(s) received no specific funding for this work.

**Competing interests:** None of the authors have any competing interests.

of students' workplace type, target age group and work seniority, but with greater improvements among those in non-problem focused role types and with less prior trauma training.

## Implications

Overall, TANDEM shows promise as a tool for supporting the implementation, evaluation, and development of trauma competence programs in health and social care contexts. Future developments should test TANDEM across diverse educational, professional, and cultural groups, increase variance in the Readiness domain, refine and validate the domains, and validate the instrument against independent, performance-based competence measures.

## 1. Background

A large body of research continues to demonstrate the negative effects of early adverse experiences on mental and somatic health, calling for preventive but also mitigating interventions [1,2]. In Norway, as in other countries, substantial public resources have been invested to strengthen "trauma competence" within health and care services. The aim has been to enhance these services' capacity to meet individuals with traumatic experiences in a way that promotes health and development. Moreover, a series of governmental action and strategic plans have been directed towards preventing violence, abuse, or neglect during childhood, as well as promoting evidence-based interventions for vulnerable groups [3].

In the Norwegian public model, five regional Resource Centers on Violence, Traumatic Stress, and Suicide Prevention (RVTSs) are assigned responsibility to promote trauma competence among service providers. Their training programs span from courses in evidence-based trauma therapy in specialized health services to long-term competence and capacity building in "trauma-informed care" (TIC) for elementary schools or child welfare institutions [4,5].

There has also been an articulate national goal to integrate trauma competence building in basic and advanced education within the higher education sector [6]. Traditionally, courses covering trauma competencies have been designed for clinical psychology students, with a focus on diagnosis and psychotherapy, although graduate-level training in trauma psychology appears to be inconsistent and sparse, in an international context [7]. Teaching trauma competencies from a public health perspective has been even less common for other mental health professionals, including psychiatric nurses [8]. At the Department of Nursing and Health Promotion (SHA) at the Oslo Metropolitan University (OsloMet), a 10-credit course on "Mental Health, Trauma and Crisis in a Life Course Perspective" has been part of the Interdisciplinary Advanced Program and Master's Program in Mental Health, Substance Use, and Addiction for many years [9].

Norwegian competence centers, such as the RVTSs, as well as OsloMet and other higher education institutions, are facing increased demands from authorities to study

and evaluate the effects of their educations and competence-building programs [6,10]. This is to ensure that the substantial public investment returns in actual increased competence among service providers.

Implementation of competency-based education involves the identification of competencies, designing of curricula, and teaching programs that are mapped to specific competencies [11]. Internationally, it has been recognized that systematic research on the effects of trauma competence programs requires standards for what the teaching should include. Several major initiatives, particularly in the U.S., have addressed this need. These include the work of the New Haven Trauma Competency Group [12,13], whose guidelines have been implemented across health and social care sectors, focusing on trauma awareness, self-reflection, and cultural sensitivity [14,15]. The National Child Traumatic Stress Network (NCTSN) offers a core curriculum [16–18] that provides structured, evidence-based education on childhood trauma, widely implemented in clinical and educational settings. Also, the Substance Abuse and Mental Health Services Administration (SAMHSA) has developed trauma-informed care principles that are integrated into public health services across the U.S., shaping organizational and policy-level practices [19–21].

However, researching the effectiveness of trauma competence programs or educational initiatives (hereafter jointly referred to as TCP) also depends on reliable and valid measures or assessment tools to evaluate students' or participants' learning outcomes. Establishing such international measures is complicated by substantial variation across countries in professionals' competence and educational levels, national legislation, and – crucially – the organization of support systems [22]. From 2018 to 2019, an expert group representing eight Norwegian competence centers assessed whether existing international assessment tools for trauma competence could, through translation, be adapted for use in a Norwegian context [22]. The aim was to find a brief instrument capturing the core competencies required for providing trauma-informed care, applicable across professions and service providers.

The group identified few instruments meeting these criteria. Several existing tools primarily target organizational readiness, such as the Trauma-Informed Care Organizational Readiness Tool (TIC-OR) [23] or the Trauma System Readiness Tool [24], or they measure the degree of implementation of trauma competence within the system, such as the TICOMETER [25]. Other instruments focus primarily on the human costs that working in this field can impose on professionals, such as the Secondary Traumatic Stress Scale (STSS) [26] and the Professional Quality of Life (ProQOL) Scale [27]. While the target groups for the relevant TCPs in Norway are highly interdisciplinary and cross-sectoral, some existing tools are designed for more narrowly defined contexts, such as schools or mental health facilities – for example, the Trauma-Informed School Practices Survey [11] – or they address specific aspects of system change, such as the Trauma-Informed System Change Instrument (TISCI) [28]. The Trauma-Informed Practice (TIP) Scales have a relevant format and scope, but assess how trauma-informed an environment is from the perspective of users [29].

The instrument that was considered the most relevant was the Attitudes Related to Trauma-Informed Care (ARTIC) scale [30], which measures professionals' attitudes toward trauma-informed care and the importance of supportive environments. It is adaptable across settings like healthcare, education, and social services. There have been no previous attempts to apply ARTIC in a Norwegian context, and based on a careful review, it was evaluated that a translated version would not serve the intended purpose. The scale was considered too basic for the competence level within Norwegian services, and it was evident that it was developed for contexts that do not reflect the total ban on corporal punishment mandated by Norwegian legislation [22].

## 1.1. TANDEM – A measure of trauma competence adapted to the Norwegian context

Based on the insights from this foundational work, in the period of 2019–2021, the same expert group developed the standardized instrument TANDEM (Trauma and Development Education Monitor) as a measure of trauma competence adapted to the Norwegian multidisciplinary context. The intention was to create an instrument serving a threefold purpose: 1) to measure the learning outcomes of TCPs, 2) to enable research on factors predicting these outcomes, and 3) to function as a guide for topics that TCPs should cover.

 

TANDEM includes 55 test items spread across six main domains: Readiness, Agency, Reflexivity, Knowledge, Skills, and Work Culture. The Knowledge, Skills, and Work Culture domains each contain two subdomains (see Method, below), bringing the total to nine domains. Note that the Skills domain was named "Practice" in the first publication [22] but was renamed at a later stage. To enable investigation of factors influencing learning outcomes, additional sections record background information on participants and the scope and content of the relevant TCP.

The structure and dimensions of TANDEM align with the areas of trauma competence developed through the initiatives of the New Haven Trauma Competency Group, NCTSN, and SAMHSA (see above). Perhaps most directly, it has been informed by SAMHSA's six core principles: Safety; Trustworthiness and transparency; Peer support; Collaboration and mutuality; Empowerment, voice and choice; Cultural, historical, and gender issues [19–21]. Similarities with the ARTIC scale are evident in TANDEM's assessment of respondents' perceived self-efficacy, their understanding of the relationship between trauma experiences and challenging behavior, their understanding of how such behavior should be addressed, as well as in the instrument's focus on developmentally supportive system and environmental variables.

All items of TANDEM, as well as the development process, are displayed in a validation study published in 2021 [22]. The study investigated the content, conceptual and criterion validity of TANDEM in a sample of 415 professionals from a broad range of Norwegian health and social services [22]. All domains and subdomains showed good scale reliability, and each also corresponded with a one-dimensional model. The instrument also distinguished well between respondents with lower versus higher levels of trauma competence and demonstrated specificity as a measure of trauma competence compared to more general competence.

How TANDEM performs as a pre–post measure remains to be scientifically tested. Adhering to best-practice measurement science and the COSMIN reporting guidelines [31], a critical aspect to examine is an instrument's responsiveness, or sensitivity to change – understood as its ability to detect meaningful changes over time [32–34]. In our context, this implies meaningful changes in respondents' TANDEM scores from before to after participating in a TCP. Given that the curriculum of the course reflects the topics assessed by TANDEM, the meaningful change to detect would be an enhancement in scores from the pre-test to the post-test. However, if, for instance, the instrument leads respondents to overestimate their competence before the course, and then, due to insights gained from the course, yields more modest or realistic scores after, this would cast doubts on the instrument's validity as a longitudinal measure of learning outcome (cf. the "Dunning-Kruger effect", [35, 36]), which would result in an underestimation of the true change.

As a collaboration between RVTS West and SHA at OsloMet, this study investigates whether TANDEM can detect changes in learning outcomes from TCPs, using data from four student cohorts who completed the survey in the TCP on Mental Health, Trauma, and Crisis in a Life Course Perspective. This overarching research question is specified into two directional hypotheses:

1. Participants' TANDEM scores are expected to increase from pre- to post-TCP, reflecting measurable improvements in trauma competence.

2. Participants' characteristics will predict level and change TANDEM scores, with greater improvements expected among those with less prior training.

## 2. Methods

### 2.1. Measures

*The TANDEM instrument:* All 55 test items in TANDEM are scored on a six-point ordinal scale, ranging from 'To a small degree' (1) to 'To a great degree' (6). The instrument is presented in its entirety in the article by Nordanger and colleagues cited above [22]. For the purpose of the current study, the analyses of changes primarily deal with the main domains, meaning that the subdomains within Knowledge, Skills, and Work Culture are treated collectively. The same test items are administered both before and after a TCP. The distribution of test items across the domains is as follows:

1. *Readiness – 3 items*. Example: "I am in favor of using such an approach in my line of work". Internal consistency reliability (Cronbach's α) for the two points of measurement was.63 and.62.

2. *Agency – 5 items*. Example: "I feel that I am able to help them". Cronbach's α was estimated to be.74 and.73 (pre- and post-test).

3. *Reflexivity – 6 items*. Example: "I have thought about what may be perceived as threatening in my way of being". Cronbach's α was.74 and.74 (pre- and post-test).

4. *Knowledge – 14 items*, divided into the sub-domains Thematic knowledge and Conceptual knowledge. Example from Thematic knowledge: "I have knowledge of how trauma may result in both too high, and too low activation levels"; Example from Conceptual knowledge: "I am familiar with the concept of dissociation".

5. *Skills – 15 items*, divided into the subdomains Supportive skills and Situational skills. Example from Supportive skills: "I feel confident on how to help them understand why they react the way they do". Example from situational skills: "I feel confident on how to meet them if they act out verbally". As noted above, this domain was named "Practice" in the original validation study [22].

6. *Work culture – 12 items*, divided into the subdomains User supportive culture and Colleague supportive culture. Example from User supportive culture: "We are conscious of how we talk about them amongst ourselves". Example from Colleague supportive culture: "We are open about situations we find difficult".

TANDEM was initially conceptualized as a reflective measurement model [37], where the overall instrument captures an underlying construct, trauma competence, and each domain reflects specific aspects of this construct. The original cross-sectional validation study of the instrument [22] established a robust factor structure, supporting the assumption that domain scores reflect distinct but related constructs. In reflective models, item responses are driven by the latent trait and are typically correlated, supported by inter-item correlations, internal consistency, and factor analysis. In contrast, formative models define the construct through its indicators, which need not be correlated [38]. The choice between these models is primarily theoretical, as empirical data alone cannot determine model type. This distinction is crucial when computing total scores across domains. A thorough conceptual review of TANDEM suggests that the first three domains, Readiness, Agency, and Reflexivity, are best understood as reflective constructs, while the remaining domains, Knowledge, Skills, and Work Culture, are more appropriately understood as formative constructs, as they are defined by their items rather than aiming to reflect an underlying latent trait. TANDEM is thus best viewed as a hybrid instrument combining both reflective and formative dimensions.

*Predictor variables*: The pre-survey includes an addendum collecting respondent information, such as *workplace type, target age group, role type, work seniority, level of basic education, and prior trauma training*. The post-survey includes an addendum gathering details about the specific TCP, including its *type, scope, content, and organizational anchoring*. The current study also investigated the relative influence of participant background characteristics on learning outcomes (see research question, above). However, here, we did not test the influence of respondents' level of basic education, since there was no variation over this variable (almost all had 3–4 years of professional education). Response categories of the included variables are:

• *Workplace type*: A list of 17 of the most common services within the Norwegian health, social and educational sectors

• *Target age group:* 1 = Children and youth; 2 = Adults; 3 = Both children/youth and adults

• *Role type*: 1 = My work is specifically targeting those who present social and emotional problems; 2 = I work with a broader population group, where I also meet those who present such problems; 3 = I work with competence development among other professionals; 4 = I have an administrative or management function; 5 = I have a more indirect relation to those who present such problems.

- *Work seniority*: A continuous scale from 'Less than a year' to 'More than 30 years.

- *Prior trauma training:* 1 = 'Very little'; 2 = 'A little'; 3 = 'Some'; 4 = 'Quite a bit', 5 = 'A lot' and 6 = 'Very much'.

To create analytically coherent groups suitable for statistical comparison, the '*Target age group*' variable was dichotomized by differentiating those who worked with children and youth in any capacity (response categories 1 and 3, above) from those who worked exclusively with adults, while the '*Workplace type*' variable was dichotomized into specialized versus non-specialized services (i.e., general or municipal health and social services). Furthermore, 4.9% of respondents who had no direct contact with users (response categories 3–5) were excluded from the "*Role type*" variable. Consequently, the Role type variable was dichotomized into "problem-focused" versus "non-problem-focused" roles (i.e., those specifically targeting clients with problems versus those serving a broader population group).

## 2.2. Sample

The sample was drawn from a total cohort of 500 students who completed the course on Mental Health, Trauma and Crisis in a Life Course Perspective, at OsloMet, in January 2021 (121 students), January 2022 (110 students), January 2023 (108 students) and January 2024 (161 students), respectively. Of these, 370 students (n2021 = 97; n2022 = 98; n2023 = 94; n2024 = 81) submitted responses on the TANDEM surveys, representing a response rate of 74%.

Due to the requirements of the study program, students had to have 1–2 years of work experience before admittance. Students were both part-time (56.5%) and fulltime students (43.5%) and a majority had health science education (70% nurses), while the rest (30%) were social welfare studies graduates. Three out of four (75%) students reported that they attended the course as the sole representatives of their workplace, while 5% reported that they were accompanied by all or almost all of their colleagues. As shown in Table 1, the largest groups in the sample worked within specialized health services, municipal health services and municipal social services.

Most of the respondents (75.1%) worked primarily with adults, 10.9% primarily with children or adolescents, and 14.2% with both age groups. For the largest group (64.1%) their work was directed towards a general target population, while 31.7% worked exclusively with users with social and emotional problems. The largest proportion of students had 3–4 years of experience (33.0%), while 18.0% had 1–2 years of experience. The vast majority (93.3%) had three or five years of professional education from before. Confer Table 2 for detailed information.

When it comes to prior training in the field of trauma, 44.4% (138) of those who responded to this item (n = 311) reported 'Very little', 35.4% (110) reported 'A little', 12.2% (38) reported 'Some', 5.1% (16) reported 'Quite a bit', and 2.9% (9) reported 'A lot'. None of the respondents reported 'Very much' prior training.

**Table 1. Sample distribution over workplace type.**

| Service | n | % |
|---|---|---|
| Specialist health services | 151 | 48.7 |
| Municipal health services | 68 | 21.9 |
| Municipal care/social services | 34 | 11.0 |
| Municipal child welfare services | 8 | 2.6 |
| Private child welfare services | 8 | 2.6 |
| Public employment services (NAV) | 8 | 2.6 |
| Government child welfare services | 4 | 1.3 |
| Criminal justice system | 1 | 0.3 |
| Police | 1 | 0.3 |
| School, children/youth | 1 | 0.3 |
| School, secondary | 1 | 0.3 |
| Other services | 25 | 2.7 |

**Table 2. Sample distribution over target age group, role type, work seniority and level of basic education.**

|  | *n* | *%* |
|---|---|---|
| Target age group |  |  |
| Children and adolescents | 33 | 10.9 |
| Adults | 232 | 75.1 |
| Both children/adolescent and adults | 44 | 14.2 |
| Role type |  |  |
| Direct work with users with social and emotional problems | 99 | 31.7 |
| Direct work with a general user population | 200 | 64.1 |
| Guiding/educating other professionals | 2 | 0.6 |
| Management/administrative function | 4 | 1.3 |
| More indirect relation | 7 | 2.2 |
| Work seniority |  |  |
| Less than 1 year | 24 | 7.8 |
| 1–2 years | 55 | 18.0 |
| 3–4 years | 101 | 33.0 |
| 5–6 years | 54 | 17.6 |
| 7–8 years | 27 | 8.8 |
| 9–10 years | 16 | 5.2 |
| 11–12 years | 11 | 3.6 |
| 13–14 years | 4 | 1.3 |
| 15 or more years | 14 | 4.4 |
| Level of basic education |  |  |
| Professional education, 3–4 years | 291 | 93.3 |
| Professional education, 5 years or more | 13 | 4.2 |
| High school* | 5 | 1.6 |
| Other education | 3 | 1.0 |

* Since admission required bachelor's degree as a minimum, these registrations are probably errors.

### 2.3. The course on "mental health, trauma and crisis in a life course perspective"

The courses on Mental Health, Trauma and Crisis in a Life Course Perspective consisted of 10 days of teaching, spread throughout three weeks, and delivered as synchronous physical lectures. The summative evaluation was a group essay based on a case, which was also discussed throughout the course. Attendance was not mandatory, except for two seminars aimed to engage in reflection and group work related to the given case. Due to Covid restrictions, the courses in 2021 and 2022 were offered in a digital format (Zoom), while the courses in 2023 and 2024 allowed for physical lectures that were also synchronously streamed.

The curriculum is informed by common guidelines for focal areas of trauma-informed care training [16,19,20], consequently also corresponding with the competence areas covered by TANDEM. In terms of interventions, environmental approaches built on principles addressed by Bath [39] and SAMHSA [19,20] were weighted more than individual psychotherapy, as this was most relevant to students' line of work. Across the weeks, cultural sensitivity in both interpretation of challenges and interventions was emphasized.

- *The first week* of the course defined and established an understanding of the concept of trauma. Further, focus was the lifelong impacts of adverse childhood experiences, including social, emotional, behavioral consequences, mental and physical health issues, posttraumatic and dissociative symptoms, and the risk of self-harm and substance abuse.

- *The second week* explored these issues from a developmental and neurodevelopmental perspective, emphasizing how a stressful and unsupportive caring environment impacts the development of self-regulation capacity, fosters dissociation as a survival strategy, that in turn may manifest as challenging behaviors and increased risks of self-harm and substance abuse. As a preventive measure, the week also included training in disclosing conversations.

- *The third week* focused on addressing these challenges, including identifying and reducing stress and triggers in the environment, de-escalating heated situations, managing dissociation and promoting coherence, providing psychoeducation, and supporting development through sensory-rich, regulating relational experiences. The core emphasis across all topics was on ourselves as the primary therapeutic tool, highlighting the need for self-reflection and a supportive, collaborative culture at work.

## 2.4. Data collection

TANDEM is web-based and takes on average around 10 minutes to complete. The survey is accessed through a QR code or a provided weblink. For the pre-survey, students in the study were provided with the QR code and weblink on the first day of teaching. The information was presented on a slide in class, shared in the Zoom chat thread for digital attenders, and made available on Canvas, the digital teaching platform used by OsloMet. Similarly, for the post-survey, the QR code and weblink were presented to students on the last day of teaching.

## 2.5. Statistical analyses

Descriptive statistics (mean, standard deviation (SD), frequencies) and internal consistency reliability (Cronbach's alpha) for the reflective measurement scales; Readiness, Agency, and Reflexivity were analyzed with SPSS version 30 [40]. For these scales, confirmatory factor analysis (CFA) was used to test longitudinal measurement invariance. These invariance models, conducted using Mplus version 8.11 [41], assess whether the necessary assumptions for analyzing change over time are met. To account for temporal dependencies, the CFA models included residual covariances (autoregressive paths) for each indicator across time points. The standard procedure is to confirm the factor model, then test for equalities in factor loadings, intercept values, and residuals within each measurement indicator over time [42]. Consequently, a free CFA model was estimated (configural invariance). Then factor loadings were constrained to be equal over time (metric, weak invariance). At the third step, the intercept values for each indicator were set to be equal over time (scalar, strong invariance). If these model constraints did not decrease the model fit, the indicator residuals were constrained to be equal over time (strict invariance). A statistically significant increase in chi-square, combined with a 0.01 decrease in CFI and a 0.015 increase in RMSEA, indicated potential problems with measurement invariance [43]. We did not test partial invariance. Substantially, the invariance analyses assess whether changes in sum scores reflect true changes in the underlying reflective construct or merely shifts at the indicator level. If individual indicators change differently over time, this may indicate that the interpretation of these indicators is evolving, rather than the latent construct itself [42]. This challenges the validity of using sum scores when analyzing changes over time. Notably, such variation at the item level can occur even when the overall factor structure and loadings appear stable. This emphasizes the importance of evaluating measurement invariance. The evaluation of model fit was based on standard goodness of fit statistics: $\chi^2$, df, p-values, comparative fit index (CFI), Tucker-Lewis Index (TLI), Root Mean Square Error of Approximation (RMSEA, point estimate, 90% confidence interval, and probability of close fit), and Root Mean Square Residuals (RMSR) [43]. These values should be non-statistically significant $\chi^2$, CFI and TLI above 0.95, RMSEA < .05, or at least < .10, and RMSR < 0.08.

Mplus was also used for estimation of Latent Growth Curve (LGC) models to examine changes in learning outcomes from pre- to post-test. These were specified according to standard guidelines [42]. The intercept factor represented pre-scores, both at mean and individual levels (variance). LGC models use all available data under the missing at random

assumption (MAR) [44] with the full information maximum likelihood (FIML). This represents an improvement compared to standard analyses under the assumption of completely at random (MCAR). Two measurement points are not sufficient to estimate a full LGC model. Consequently, the residual variances were constrained to be equal, and the slope variance fixed to zero, which also implies the covariance between intercept and slope to be zero. This equals the random intercept fixed slope linear mixed (multilevel) model [45]. Based on the LGC results, post-estimated scores were computed by taking the pre-estimated scores + estimated change. We also estimated effect sizes (ES), which were computed by dividing the estimated change on SD at pre-test. Model fit was not relevant in these restricted models.

We examined potential moderating effects of respondents' background characteristics on changes in outcome variables by adding these covariates on intercept and slope in the LGC models. The analyses were based on the sample of 370 respondents. While some relationships among domains were partially exploratory, others were hypothesis-driven. Therefore, we did not consider the entire set of analyses as purely exploratory. Given the large sample size, which implies substantial statistical power, and the 35 estimated regression coefficients, we acknowledge the potential for inflated findings. However, because these associations stem from related but distinct models addressing different hypotheses, we did not apply any correction for potential Type I errors, familywise error rate (FWER), or false discovery rate (FDR).

## 2.6. Ethics and privacy

Participation in the TANDEM survey is based on informed consent. On the opening page, key information about the survey is provided, along with a link to a comprehensive consent form. Participants consent to participate based on confirming that they have read and understood the information. The survey ensures respondent anonymity. Only non-sensitive personal information such as educational background, experience, and organizational affiliation is requested, while gender, age, IP addresses or any other information that could identify respondents is not. The survey was developed and set up in consultation with privacy authorities at OsloMet and Helse-Bergen HF, where RVTS West is affiliated.

## 3. Results

### 3.1. Measurements invariance

The model fit results from the CFA measurement invariance analyses are presented in S1 Table and the standardized factor loadings for the freely estimated models in S2 Table. For the Readiness dimension, configural invariance was supported, indicating that the factor structure remained stable over time. Metric invariance was also supported, as factor loadings were equivalent across time points. Although constraining item intercepts (scalar invariance) led to a non-significant model fit, the chi-square difference test showed a statistically significant increase in chi-square. Although model fit deteriorated, the goodness of fit indices still supported strong invariance for the Readiness construct. For the Agency dimension, both configural and metric invariance were supported, but the statistically significant increase in chi-square, lower levels in CFI and TLI, and increase in RMSEA showed reduced support for scalar invariance. Similarly, the results for the Reflexivity dimension showed support for configural and metric invariance, but not scalar invariance. These results indicate weak invariance for both Agency and Reflexivity dimensions. No residual covariance had to be included in the models in order to improve fit-measures.

### 3.2. Changes in scores from pre to post-test

The observed and estimated mean and SD of pre- and post-test, as well as LGC results, are presented in Table 3. The results showed statistically significant improvement from pre-test to post-test, indicating a substantial enhancement in participants' overall reported competence. In addition, scores on all sub-domains (Agency, Readiness, Reflexivity, Knowledge, Skills and Work culture) were found to increase over time. The ES results showed strongest improvement in Knowledge, total scores, and Skills, all with strong effects.

**Table 3. Descriptive information for Pre- and Post-Test and Latent Growth Curve Model (LGCM) results for learning outcomes, total and per domain.**

| | Pre-Test | | | Post-test | | | Change | | | | | | |
|---|---|---|---|---|---|---|---|---|---|---|---|---|---|
| | *n* | *M* | *SD* | *n* | *M*[a] | *SD*[b] | Δ | *95%CI* | | *P* | *ES* | *95%CI* | |
| Total score | 295 | 200.46 | 38.79 | 214 | 241.50 | 27.93 | | | | | | | |
| LGCM | 347 | 205.88 | 27.54 | | 243.71 | 34.82 | 37.85 | 33.84 | 41.86 | <.001 | 1.37 | 1.17 | 1.58 |
| Readiness | 315 | 15.32 | 2.81 | 238 | 15.88 | 2.51 | | | | | | | |
| LGCM | 368 | 15.33 | 1.78 | | 15.82 | 2.69 | 0.49 | 0.13 | 0.85 | .008 | 0.28 | 0.06 | 0.49 |
| Agency | 315 | 21.44 | 3.91 | 237 | 22.46 | 3.63 | | | | | | | |
| LGCM | 366 | 21.37 | 2.94 | | 22.63 | 3.81 | 1.26 | 0.81 | 1.71 | <.001 | 0.43 | 0.27 | 0.59 |
| Reflexivity | 310 | 25.75 | 5.05 | 235 | 27.70 | 4.60 | | | | | | | |
| LGCM | 365 | 25.65 | 3.20 | | 27.78 | 4.87 | 2.14 | 1.46 | 2.81 | <.001 | 0.67 | 0.44 | 0.89 |
| Knowledge | 310 | 48.00 | 14.58 | 233 | 64.77 | 8.87 | | | | | | | |
| LGCM | 365 | 47.90 | 8.34 | | 65.12 | 12.49 | 17.23 | 15.63 | 18.82 | <.001 | 2.07 | 1.70 | 2.43 |
| Skills | 311 | 50.08 | 14.10 | 232 | 61.93 | 10.94 | | | | | | | |
| LGCM | 362 | 49.92 | 10.14 | | 62.76 | 12.90 | 12.84 | 11.39 | 14.30 | <.001 | 1.27 | 1.06 | 1.48 |
| Work Culture | 306 | 45.90 | 11.77 | 233 | 48.09 | 10.96 | | | | | | | |
| LGCM | 359 | 45.65 | 10.00 | | 48.97 | 11.54 | 3.32 | 2.19 | 4.44 | <.001 | 0.33 | 0.22 | 0.44 |

Δ: LGCM Difference: Post – Pre scores, 95%CI: 95% confidence interval.

ES: Effect size based on estimated Δ / estimated Pre-test SD (from LGCM).

[a]Post-Test is estimated by computing Pre-Test minus latent change from LGCM.

[b]Observed SD estimated by taking the square root of the variances of intercept and residuals.

### 3.3. Correlations between respondent background characteristics, baseline scores, and learning outcomes

Before including respondent background variables as potential predictors of learning outcome in the regression analyses, correlations of predictors and correlations between the predictors and baseline and learning outcome scores were analyzed. The analyses showed statistically significant higher learning outcomes among respondents with less prior trauma training and lower baseline scores, as well as among respondents in non-problem focused roles (i.e., not exclusively serving clients presenting problems, see above) (Table 4).

Respondents in non-problem-focused roles had significantly lower baseline scores compared to those in problem-focused roles. Lower baseline scores were significantly more common among respondents with less prior trauma training. Less prior trauma training, in turn, was significantly more common among respondents working exclusively with adults in non-specialized services, and among those with shorter work experience. The correlation values indicate that the variables represent sufficiently distinct phenomena to be included separately in the regression analyses.

### 3.4. Predictors of learning outcomes

When respondents' background characteristics were included as predictors in the conditional LGC analyses, total learning outcome was greater among those with less prior training and among those in non-problem-focused roles (Table 5). However, these respondents also showed lower pre-test total scores (b = 18.22, β = 0.58 and b = −15.43, β = −0.23; both p < .001, respectively). In addition, Table 5 shows that working in specialized health services was associated with greater improvements in Agency, but these also had lower pre-test levels in this domain (b = −1.32, β = −0.22, p = .002). In addition, higher pre-test Agency levels were found among those in non-problem-focused roles (b = −1.11, β = −0.17, p = .018) and among those with higher pre-test levels of prior trauma training (b = 0.69, β = 0.23, p = .006). Despite no statistical associations between the predictors and improvements in Readiness, respondents in non-problem-focused compared

**Table 4. Correlation matrix of correlations between respondent background characteristics, baseline score and learning outcome (Pearson's r) (N = 357).**

|  | 1 | 2 | 3 | 4 | 5 | 6 |
|---|---|---|---|---|---|---|
| Baseline total score (1) |  |  |  |  |  |  |
| Workplace type [a] (2) | .03 |  |  |  |  |  |
| Target age group [b] (3) | −.19* | .17* |  |  |  |  |
| Role Type [c] (4) | −.37** | .07 | .22** |  |  |  |
| Work seniority [d] (5) | .19* | −.02 | −.01 | .07 |  |  |
| Prior trauma training (6) | .64** | −.05 | −.25** | −.29** | .24** |  |
| Learning outcome (7) | −.63** | .06 | .08 | .42* | −.07 | −.66** |

* = $p <$.05 ** = $p <$.001

[1,7] "Baseline total score" and "Learning outcome" taken from the LGCM intercept and slope factors.

[a]Working in specialized health services; Reference group: working in general/municipal services

[b]Working with adults; Reference group: working with children and youth

[c]Working with a broader population group; Reference group: working with those with problems

[d]Work seniority from "less than a year" to "more than 30 years".

to problem-focused roles, showed lower pre-test levels (b = −0.92, β = −0.24, p = .007). Less improvement in Reflexivity was found among those with more prior trauma training, but such higher levels of prior training were also associated with higher pre-test Reflexivity levels (b = 0.76, β = 0.15, p = .012). In addition, those respondents in non-problem-focused roles reported lower levels of Reflexivity (b = −1.82, β = −0.17, p = .004). Stronger improvement in Knowledge was found among those with less prior trauma training (Table 5). However, such lower levels were also related lower pre-test levels in Knowledge (b = 7.79, β = 0.72, p < .001), as well as having non-problem focused versus problem-focused roles (b = −3.61, β = −0.16, p = .029). Increasing Skills were found to be stronger among respondents with less prior trauma training and among those working in non-specialized services. At the pre-test level, lower levels in Skills were associated with lower levels of prior trauma training (b = 5.76, β = 0.52, p < .001), working within general/municipal services (b = 2.83, β = 0.13, p = .040), and having a non-problem-focused role (b = −4.48, β = −0.19, p = .006). We found no predictors to be related to changes in Work culture, in spite of findings showing lower pre-test levels among respondents serving the general population (b = −3.13, β = −0.14, p = .035) and having lower levels of prior trauma training (b = 3.06, β = 0.30, p < .001).

## 4. Discussion

The study aimed to evaluate whether the TANDEM instrument is sensitive to meaningful changes [32–34] in trauma competence from pre- to post-intervention in a TCP, hypothesizing that it would reflect measurable improvement, with less prior trauma training being the strongest predictor of both score level and change. Such sensitivity was demonstrated among the four cohorts of students who followed the course on Mental Health, Trauma and Crisis in a Life Course Perspective at OsloMet in the period of 2021–2024. As hypothesized, a statistically significant increase in sum scores from pre- to post-program was found. The increase applied to both the total scale and all individual domains, though, given the invariance findings, the observed changes in Agency and Reflexivity cannot be interpreted as evidence of genuine gains in competence in these domains. Overall, the detected pre–post changes appear meaningful, as the course curriculum aligns closely with TANDEM's assessment foci.

The overall effect size detected by TANDEM (d = 1.37) appears to fall within a similar range to ARTIC, which may be the most relevant instrument for comparison (see Introduction). Among the few studies we identified that have examined ARTIC as a pre–post measure of trauma training outcomes, reported effect sizes range from approximately d = 0.5 among multi-disciplinary clinical staff [46] to d = 1.35 among school personnel [47]. We recognize, however, that for TANDEM,

**Table 5. Relations between background characteristics and learning outcome (N = 357).**

| | Total | | | Readiness | | | Agency | | | Reflexivity | | | Knowledge | | | Skills | | | Work culture | | |
|---|---|---|---|---|---|---|---|---|---|---|---|---|---|---|---|---|---|---|---|---|---|
| | b | β | p | b | β | p | b | β | p | b | β | p | b | β | p | b | β | p | b | β | p |
| Intercept | 50.23 | | | −0.83 | | | 0.23 | | | 3.87 | | | 29.06 | | | 14.56 | | | 3.67 | | |
| Workplace type [a] | 0.44 | 0.02 | .913 | 0.39 | 0.46 | .344 | 1.04 | 0.72 | .040 | 0.81 | 0.08 | .277 | −1.08 | −0.08 | .475 | −0.61 | −0.08 | .692 | 0.21 | 0.06 | .859 |
| Target age group [b] | −6.61 | −0.21 | .137 | −0.07 | −0.07 | .871 | −0.09 | −0.05 | .879 | −0.86 | −0.08 | .382 | −2.24 | −0.15 | .180 | −0.94 | −0.11 | .587 | −2.14 | −0.54 | .130 |
| Role type [c] | 10.71 | 0.38 | .008 | 0.02 | 0.02 | .968 | 0.15 | 0.10 | .782 | 0.37 | 0.04 | .644 | 2.64 | 0.20 | .113 | 3.78 | 0.47 | .018 | 2.19 | 0.60 | .060 |
| Work seniority [d] | 0.58 | 0.09 | .561 | 0.07 | 0.33 | .503 | 0.07 | 0.19 | .564 | 0.06 | 0.03 | .762 | 0.18 | 0.06 | .605 | 0.00 | 0.00 | .995 | 0.34 | 0.42 | .318 |
| Prior Trauma Training | −11.72 | −0.88 | <.001 | 0.33 | 0.77 | .126 | −0.47 | −0.66 | .087 | −1.18 | −0.24 | .006 | −6.15 | −0.97 | <.001 | −2.98 | −0.79 | <.001 | −1.02 | −0.60 | .156 |
| $R^2$ * | 27.2% | | | 2.1% | | | 4.1% | | | 6.3% | | | 33.2% | | | 16.0% | | | 5.7% | | |

b: unstandardized regression coefficient; β: standardized regression coefficient.

[a] Working in specialized health services; Reference group: working in general/municipal services.

[b] Working with adults; Reference group: working with children and youth.

[c] Working with a broader population group; Reference group: working with those with problems.

[d] Work seniority from "less than a year" to "more than 30 years".

* To estimate explained variance the fixed slope variance was freed and residuals set to zero.

the effect sizes were somewhat unevenly distributed across domains, with the far greatest pre–post changes observed in Knowledge and Skills domains. This, however, also appears to represent a meaningful profile across domains, as relatively greater gains in these areas are to be expected in the context of a three-week, university-level course. Unlike longer workplace TCPs, such a course offers limited opportunity to foster self-reflexivity and work culture through staff supervision.

Following the same logic, although the effect size was small, the statistically significant learning outcome on Work Culture was somewhat surprising, given that 75% of the sample attended as the sole representatives of their workplaces (see Sample, above). Possibly, this finding reflects shifts in respondents' perceptions of their workplaces rather than actual cultural change, although real changes cannot be ruled out. Many students work while studying and are expected to share what they learn with colleagues; moreover, 5% of students attended together with all or nearly all of their closest colleagues.

As readiness is a key factor in shifting organizations toward a trauma-informed approach [21] and the focus of several assessment tools [23,24], the small effect size on this domain may warrant attention. This finding may first and foremost reflect the group's high baseline Readiness scores (15.3/18), leaving limited room for improvement (ceiling effect). However, as respondents are representative of TANDEM's target groups, limited variance may in itself be a concern, calling for revised or additional items capturing a broader range of readiness.

As hypothesized, prior trauma training emerged as the main predictor of learning outcomes, with the greatest pre–post improvement observed among participants with lower levels of prior training. Still, this finding should be interpreted with some caution, as respondents with particularly low or high scores will naturally exhibit some "regression to the mean" [48] upon retesting. However, the general increase across all participants may rule out this phenomenon as the sole explanation, and overall, the results can likely be interpreted as an indication of TANDEM's ability to detect learning outcomes regardless of workplace type, target age group and work seniority. In this sense, TANDEM shows promise in handling diverse contexts, target groups, and TCP formats engaged by competence centers and higher education institutions, fulfilling its intended purpose.

Nevertheless, according to our findings, TANDEM may not be equally applicable across all groups. Those with less prior trauma training were more often respondents who worked primarily with adults in general rather than specialized services. This is consistent with the historical view of trauma competence as a relatively specialized skill, and thus less frequently offered to general services. Additionally, in Norway, investment in building the type of trauma competence measured by TANDEM has likely been greater in services for children and adolescents. For example, substantial resources have been devoted to implementing trauma-informed care across schools and child welfare services [4,5,49–51].

This means that some caution may be needed when TANDEM is administered to groups with considerable prior trauma training, as will more likely be the case in specialized child-centered services. Of course, steeper learning curves among those with little prior training do not necessarily indicate anything about the instrument's sensitivity; it is equally plausible to see this as a reflection of an inherent nature of learning, where those with more to learn tend to learn more. Nevertheless, the finding suggests that if a TCP is designed using the domains of TANDEM as a guide, professionals with higher baseline scores may not reach their full learning potential.

If such a tendency exists, it needs not to be viewed as a problem, but rather as being consistent with the instrument's primary purpose. The purpose is to measure basic trauma competence, and to facilitate enhancing the level of such competence across sectors and services. Despite some uncertainty regarding the mechanisms underlying the instability observed in the Agency and Reflexivity domains – where overestimation of self-competence on individual items at pre-test may have played a role – the overall impression is that TANDEM avoids a "Dunning–Kruger effect" [35,36]; that is, it does not appear to prompt overestimation of pre-program competence, with a drop in post-program scores due to a more modest self-assessment evoked by the course. This finding bodes well for the future application of TANDEM in TCPs and in implementation research.

## 4.1. Conceptualization of measurement model

The domains Readiness, Agency, and Reflexivity were developed and analyzed as reflective constructs. The longitudinal invariance analyses provided support for strong measurement invariance only for Readiness. For Agency and Reflexivity, factor loadings were equal across time, but mean structures were not invariant. This means that observed pre–post changes cannot be confidently interpreted as true changes in the underlying constructs, as differences in how participants understood or responded to individual indicators over time may also have contributed to the observed differences. Such patterns may, for example, reflect increased insight or more nuanced interpretations of phrasing of items resulting from the educational intervention, and should therefore be interpreted with caution. On the one hand, seeing something new in items post program may even be a desired part of a learning process, reflecting a professional maturation fostered by the TPC. Still, from the perspective of instrument precision – ensuring clarity about what is captured – there may be a need to reconsider the dimensionality of these parts of the instrument.

The domains Knowledge, Skills, and Work Culture were, by contrast, conceptualized as formative. Here, the indicators constitute the construct rather than being manifestations of a common underlying factor, making internal consistency reliability, confirmatory factor and measurement invariance analyses irrelevant, as these are based on the reflective measurement assumption. As outlined in the Method section, the distinction between reflective and formative dimensions in TANDEM was based on a thorough review of the conceptual ideas underlying each domain. Still, it can be debated whether some of the domains defined as formative may contain reflective elements, which would have implications for how the hybrid measurement model is understood and for the interpretation of results within these domains.

## 4.2. Limitations and next steps

A limitation relates to the context and characteristics of the sample. The sample consisted solely of students from Oslo Metropolitan University, with an overrepresentation of participants holding three to four years of vocational education and working primarily with adults in the healthcare sector. This distribution reflects the predominance of nursing students within the Faculty of Health Sciences (approximately 70%, see above). A natural next step would be to include a more diverse group of participants representing a broader range of educational and professional backgrounds. Ideally, this should encompass professionals outside the health sector as well as individuals without formal professional training. Ensuring regional and cultural variation should also be a priority, given Norway's multicultural composition across both geographical regions and service structures.

It should also be noted that the first two cohorts in this study received their teaching during the exceptional circumstances of the COVID-19 pandemic. It remains unclear how the mode of instruction – digital versus in-person – may have influenced the results. The absence of direct comparisons between learning outcomes across these formats constitutes an additional limitation and a relevant topic for future research.

In the next phase, construct and cross-cultural validation studies, with TANDEM being tested in different professional groups and cultural contexts, are needed to consolidate the instrument's scientific foundation. The current sample is likely subject to some selection bias, for example with regard to participants' interests and attitudes. Although participation in the course was mandatory, students had voluntarily enrolled in the broader educational program, suggesting a pre-existing openness toward the subject matter. This is also evident in the high baseline Readiness scores. Such an attitude may also have influenced scores in other domains. Consequently, future studies should examine how TANDEM performs among groups that may approach the topic with greater skepticism or hesitation.

Another important next step would be to expand the TANDEM portal to include a follow-up measurement, for instance after six months – something which is already underway. At present, uncertainties remain regarding the extent to which sustained competence is captured, as data were collected immediately after course completion, when the material was still fresh in participants' minds.

In future studies, to gain better insight into possible shifts in participants' perspectives over time, we will consider exploring their retrospective evaluations by including "then–retro" ratings at the post-assessment point [52]. Furthermore, when dimensions do not show measurement invariance, analyzing changes at indicators rather than at domain scale level may add clarity to what pre–post changes mean. Lastly, in accordance with COSMIN standards for criterion validity [31], future studies should compare TANDEM scores with performance-based measures and criteria reflecting service-user perspectives. Such comparisons would provide valuable insight into how TANDEM scores translate into tangible improvements in the actual quality of services provided.

## 5. Conclusion

TANDEM shows promise as a tool for supporting the implementation, evaluation, and development of trauma competence programs in health and social care contexts. The instrument captured changes in scores from before to after the TPC in a way that appears meaningful against the background of the course curriculum. Future developments should test TANDEM across diverse educational, professional, and cultural contexts, increase variance in the Readiness domain, investigate interpretative changes in the Agency and Reflexivity domains, and validate the instrument against independent, performance-based competence measures.

## Supporting information

**S1 Table. Measurement invariance results.**
(DOCX)

**S2 Table. Standardized factor loadings for readiness, agency, and reflexivity.**
(DOCX)

**S1 File. Codebook TANDEM sensitivity.**
(PDF)

## Author contributions

**Conceptualization:** Dag Ø. Nordanger, Alina Coman.

**Data curation:** Rolf Gjestad, Dag Ø. Nordanger.

**Formal analysis:** Rolf Gjestad, Dag Ø. Nordanger.

**Investigation:** Dag Ø. Nordanger, Alina Coman.

**Methodology:** Rolf Gjestad, Dag Ø. Nordanger, Alina Coman.

**Project administration:** Anca Maria Yttri.

**Resources:** Anca Maria Yttri.

**Supervision:** Anca Maria Yttri.

**Writing – original draft:** Rolf Gjestad, Dag Ø. Nordanger, Alina Coman, Anca Maria Yttri.

**Writing – review & editing:** Rolf Gjestad, Dag Ø. Nordanger, Alina Coman.

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
