## [Decision Letter · Decision Letter 0]

3 Jan 2025

PONE-D-24-48365The sensitivity of TANDEM – a new measure of trauma competencePLOS ONE

Dear Dr. Nordanger,

Thank you for submitting your manuscript to PLOS ONE. After careful consideration, we feel that it has merit but does not fully meet PLOS ONE’s publication criteria as it currently stands. Therefore, we invite you to submit a revised version of the manuscript that addresses the points raised during the review process. **The study effectively demonstrates the sensitivity of the TANDEM instrument in assessing trauma-informed care education, but there are areas for improvement. While pre-post changes in self-perceived competence are evident, future validation should include performance-based measures and service user perceptions to strengthen its applicability. The lack of change in the Readiness domain warrants deeper analysis, and the rationale for the additional exploratory factor analysis should be clarified. Expanding the participant pool to include diverse educational and professional backgrounds would enhance generalizability. Additionally, exploring long-term retention of knowledge and cross-cultural validation will provide a more comprehensive understanding of TANDEM's reliability and effectiveness.** 

We look forward to receiving your revised manuscript.

Kind regards,

Muhammad Shahzad Aslam, Ph.D.,M.Phil., Pharm-D

Academic Editor

PLOS ONE

**Journal Requirements:**

3. Please include your tables as part of your main manuscript and remove the individual files. Please note that supplementary tables (should remain/ be uploaded) as separate "supporting information" files.

**Additional Editor Comments:**

1-The study assesses self-perceived competence rather than actual performance.

Suggest comparing TANDEM scores with performance-based measures or service user perceptions of trauma-informed care for future validation.

2-Clarify the purpose of the additional EFA analysis and its relevance to the research question.

3-Acknowledge that students with the lowest pre-intervention scores showed the most improvement, which aligns with expectations but requires further explanation of the processes involved.

4-Explore why no significant change was observed in the Readiness domain. A deeper analysis of specific items or influencing factors (e.g., attitudes, personal readiness) is recommended.

5-Include participants from various educational and professional backgrounds to validate TANDEM's applicability across broader contexts and enhance generalizability.

6-Investigate factors such as individual motivation or personal trauma experiences that may influence learning outcomes but were not addressed in this study.

7-Assess long-term retention of knowledge and skills acquired during the course to provide a more comprehensive evaluation of TANDEM's effectiveness.

8-Conduct further validation studies in different regions or cultural contexts to confirm TANDEM’s reliability and sensitivity in diverse settings.

Reviewers' comments:

Reviewer's Responses to Questions

**Comments to the Author**

1. Is the manuscript technically sound, and do the data support the conclusions?

Reviewer #1: Yes

Reviewer #2: Yes

2. Has the statistical analysis been performed appropriately and rigorously? 

Reviewer #1: Yes

Reviewer #2: Yes

3. Have the authors made all data underlying the findings in their manuscript fully available?

Reviewer #1: Yes

Reviewer #2: Yes

4. Is the manuscript presented in an intelligible fashion and written in standard English?

Reviewer #1: Yes

Reviewer #2: Yes

5. Review Comments to the Author

**Reviewer #1:**  Review of The sensitivity of TANDEM – a new measure of trauma competence

This paper is well written. The study is straightforward and is reported clearly. The analysis is appropriate for the research question.

A very general point is that this study aimed to determine whether the TANDEM instrument [is effective] in detecting “meaningful changes”. The study reported here has demonstrated that there is a statistically significant difference in the pre post scores in students who have received a course. However, this is mainly assessing self-perceived competence. We don’t know that these changes are meaningful in terms of performance. There are two potential issues firstly that students may misperceive their own competence and secondly social desirability bias. They have just received a course and are then asked whether they have met the learning outcomes (which is essentially what TANDEM is asking them). Many students when asked will say they have met learning outcomes. Students may not accurately estimate their ability to undertake activities.

The original study (referenced as 24) which established criterion validity, did so by comparing TANDEM scores to self-reported training/education (RMT). I think it might be worth considering that a comparison to performance as opposed to self-reported competence would be a useful future addition to the validation of TANDEM. Does TANDEM score align with service user’s perception of whether care is trauma informed?

I think also it would be useful to have more explanation of why the additional EFA analysis was conducted and what element of the research question this answered.

It is not surprising that those with the lowest scores pre intervention improved the most and processes such as ‘regression to the mean’ should be considered as an alternative explanation.

**Reviewer #2:**  The study provides valuable insights into the sensitivity of the Trauma and Development Education Monitor (TANDEM) in evaluating trauma-informed care education. The use of pre- and post-test data to assess changes in learning outcomes among students is a strong approach, and the results demonstrate that TANDEM effectively measures changes in knowledge, skills, and organizational culture, particularly for students with less prior trauma training.

Suggestions for Improvement:

1. The study mentions that no significant change was observed in the Readiness domain. It would be helpful to explore why this is the case. Perhaps a deeper analysis of the specific items within this domain could shed light on whether it was less responsive to the course or if other factors (e.g., prior attitudes, personal readiness) played a role.

2.While the study sample comes from Oslo Metropolitan University, it would be beneficial to include a more diverse group of participants from different educational backgrounds and professional sectors, as the authors mention. This could help validate TANDEM’s applicability across a broader range of contexts and enhance its generalizability.

3. It would be useful to explore additional variables, such as the role of individual motivation or personal experience with trauma, which may also affect learning outcomes but were not fully addressed in this study. Including these factors could help refine the instrument’s sensitivity.

4. The study primarily focuses on immediate pre-post changes. Future studies could explore the long-term retention of the knowledge and skills acquired during the course. This would provide a more complete picture of TANDEM's effectiveness in supporting sustained competence in trauma-informed care.

5. While the study suggests promising psychometric qualities for TANDEM, further validation studies in different regions or cultural contexts would be beneficial to confirm its reliability and sensitivity across diverse settings.

6. PLOS authors have the option to publish the peer review history of their article (what does this mean? ). If published, this will include your full peer review and any attached files.

**Do you want your identity to be public for this peer review?** For information about this choice, including consent withdrawal, please see our Privacy Policy .

Reviewer #1: No

Reviewer #2: No

---

## [Author Response · Author response to Decision Letter 1]

7 Apr 2025

Please see the attached "Response to Reviewers" document for a systematic overview of how we have responded to each of the reviewers' comments.

---

## [Decision Letter · Decision Letter 1]

24 Apr 2025

PONE-D-24-48365R1The sensitivity of TANDEM – a new measure of trauma competencePLOS ONE

Dear Dr. Nordanger,

Thank you for submitting your manuscript to PLOS ONE. After careful consideration, we feel that it has merit but does not fully meet PLOS ONE’s publication criteria as it currently stands. Therefore, we invite you to submit a revised version of the manuscript that addresses the points raised during the review process.

Clarify Cohort Size and Response RateAlign Domain Count with Nordanger et al. (Ref. 25)Specify Formative vs. Reflective Measurement ModelAddress Regression‐to‐the‐Mean Concern

We look forward to receiving your revised manuscript.

Kind regards,

Muhammad Shahzad Aslam, Ph.D.,M.Phil., Pharm-D

Academic Editor

PLOS ONE

**Additional Editor Comments:**

Recalculate and report the response rate using that exact denominator to ensure that the percentage aligns with 167 respondents.The original validation by Nordanger et al. employed nine domain scores (six main domains, three split into subdomains). Please clarify whether your use of six domains reflects aggregation of those subdomains, or else adjust your description so that the number of domains matches the validation reference.Ensure consistency between the manuscript text, your tables/figures, and the factor‐analysis results reported in Ref. 25.Explicitly state whether TANDEM is conceptualized as a reflective model (single latent construct driving domain scores) or a formative model (competence formed by distinct domain scores).In your Results or Discussion section, briefly acknowledge the mathematical coupling between baseline scores and change scores (e.g. Clifton & Clifton, 2019, Trials 20:43), and explain how your analytical approach mitigates any undue bias from regression to the mean.

Reviewers' comments:

Reviewer's Responses to Questions

**Comments to the Author**

1. If the authors have adequately addressed your comments raised in a previous round of review and you feel that this manuscript is now acceptable for publication, you may indicate that here to bypass the “Comments to the Author” section, enter your conflict of interest statement in the “Confidential to Editor” section, and submit your "Accept" recommendation.

Reviewer #1: (No Response)

Reviewer #2: All comments have been addressed

2. Is the manuscript technically sound, and do the data support the conclusions?

Reviewer #1: Yes

Reviewer #2: Yes

3. Has the statistical analysis been performed appropriately and rigorously? 

Reviewer #1: Yes

Reviewer #2: Yes

4. Have the authors made all data underlying the findings in their manuscript fully available?

Reviewer #1: No

Reviewer #2: Yes

5. Is the manuscript presented in an intelligible fashion and written in standard English?

Reviewer #1: Yes

Reviewer #2: Yes

6. Review Comments to the Author

Reviewer #1: Review of ‘The sensitivity of TANDEM - a new measure of trauma competence’

Thank you for inviting me to review the revised version of the manuscript. The authors have been very thorough in their response.

My comments are as follows:

1. I had not previously noticed that the first paragraph under ‘2.2 Sample’ says that each year an average of 90 students complete this course so over 3 years this would be 270. The response rate of 167 is reported as 74% of the cohort but 167 is 61.9% of 270. I would give the exact number of people who completed the course over the 3 years rather than the average per year.

2. The section on multiple regression analysis in the ‘2.5 Statistical analyses’ section now states the ’validation study of the instrument (25) established a robust factor structure, supporting the assumption that domain scores reflect distinct but related constructs, providing a valid foundation for the regression analyses’. However, the ref 25 Nordanger et al paper uses 9 domains in their validation analyses (tables 1-5). These are the 6 main domains but 3 are split into two subdomains and all 9 were included separately in their factor analysis as far as I can tell. I don’t think this is a big issue, but the number of domains should be consistent with the reference.

I think it would be good to be explicit about whether this was based on a formative or a reflective measurement model. That is theoretically, for a reflective model there is a single underlying latent construct of trauma competence that drives all the scores on this measure. Alternatively in a reflective model trauma competence is made up of, or formed by, competence in the different areas/domains covered by the measure. This entails that scores on the domains will not necessarily be highly correlated and therefore they can be regressed separately. I think it would be helpful to readers to be more explicit about the formative versus reflective nature of the measure under discussion.

E.g. Chang et al 2016 Comparing reflective and formative measures: New insights from relevant simulations - ScienceDirect Journal of Business Research

Volume 69, Issue 8, Pages 3177-3185

3. My comment about the regression to the mean referred to the correlation between baseline score and the learning outcome ‘Learning outcome was most strongly correlated with low baseline score (r = -.68, p < .001)’. This is because mathematically it can be shown that there is always a correlation between the baseline score and the change score. This is shown by equation 4 in this paper

Clifton, L., Clifton, D.A. The correlation between baseline score and post-intervention score, and its implications for statistical analysis. Trials 20, 43 (2019). https://doi.org/10.1186/s13063-018-3108-3

Reviewer #2: All issues have been addressed well. I recommend the acceptance of this paper. I believe readers will gain a lot from this paper.

7. PLOS authors have the option to publish the peer review history of their article (what does this mean? ). If published, this will include your full peer review and any attached files.

**Do you want your identity to be public for this peer review?** For information about this choice, including consent withdrawal, please see our Privacy Policy .

Reviewer #1: **Yes:** Catherine Best

Reviewer #2: No

---

## [Author Response · Author response to Decision Letter 2]

1 May 2025

Se the attached "Response to reviewers" letter.

---

## [Decision Letter · Decision Letter 2]

12 Jun 2025

PONE-D-24-48365R2The sensitivity of TANDEM – a new measure of trauma competencePLOS ONE

Dear Dr. Nordanger,

Thank you for submitting your manuscript to PLOS ONE. After careful consideration, we feel that it has merit but does not fully meet PLOS ONE’s publication criteria as it currently stands. Therefore, we invite you to submit a revised version of the manuscript that addresses the points raised during the review process. While the rationale for developing TANDEM and its alignment with established trauma-informed care frameworks are strengths, the reviewers request clearer, hypothesis-driven research questions; empirical justification (e.g., pilot data) for why existing tools underperform in Norway; consistent terminology; and adherence to psychometric standards (e.g., COSMIN responsiveness). Methodological improvements include explaining the lack of a control group, providing a participant flow diagram with attrition analysis, implementing robust missing-data handling (e.g., multiple imputation), reporting effect sizes with confidence intervals, avoiding dichotomization in regression, and presenting detailed EFA and measurement-invariance analyses. Additionally, they ask for enhanced data-sharing plans, visual summaries of domain-specific change (e.g., line graphs), deeper discussion of response-shift bias, comparison of effect sizes to other instruments, tempered conclusions, and alignment with STROBE/COSMIN reporting checklists.

We look forward to receiving your revised manuscript.

Kind regards,

Muhammad Shahzad Aslam, Ph.D.,M.Phil., Pharm-D

Academic Editor

PLOS ONE

Additional Editor Comments:

Include pilot or comparative data demonstrating why existing instruments (e.g., ARTIC, TICOMETER) underperform in Norway.Cite and align responsiveness evaluation with COSMIN (COnsensus‐based Standards for the selection of health Measurement Instruments) guidance.Choose one term for the training initiative—“course,” “programme,” or “TCP”—and apply it uniformly.Provide a CONSORT‐style flow diagram and an attrition analysis to identify potential bias.Report paired-samples Cohen’s d or standardized response means (SRM) with 95 % confidence intervals for all pre/post change estimates.Report standardized β coefficients with confidence intervals to convey predictor importance.

Reviewers' comments:

Reviewer's Responses to Questions

**Comments to the Author**

1. If the authors have adequately addressed your comments raised in a previous round of review and you feel that this manuscript is now acceptable for publication, you may indicate that here to bypass the “Comments to the Author” section, enter your conflict of interest statement in the “Confidential to Editor” section, and submit your "Accept" recommendation.

Reviewer #1: All comments have been addressed

Reviewer #3: (No Response)

2. Is the manuscript technically sound, and do the data support the conclusions?

Reviewer #1: Yes

Reviewer #3: Yes

3. Has the statistical analysis been performed appropriately and rigorously? 

Reviewer #1: Yes

Reviewer #3: No

4. Have the authors made all data underlying the findings in their manuscript fully available?

Reviewer #1: No

Reviewer #3: No

5. Is the manuscript presented in an intelligible fashion and written in standard English?

Reviewer #1: Yes

Reviewer #3: Yes

6. Review Comments to the Author

Reviewer #1: Thank you to the authors for carefully addressing my previous comments. I am happy that the manuscript is ready for publicaiton.

Reviewer #3: The manuscript opens with a clear and persuasive rationale for examining whether the TANDEM instrument can detect learning gains in trauma-competence programmes. The authors effectively frame the need for a Norwegian-specific tool by contrasting TANDEM with established international scales such as ARTIC and TICOMETER and by highlighting national investments in “trauma competence.” The background on TANDEM’s development, and its anchoring in frameworks like SAMHSA’s core principles, is concise and informative.

Strengths

A well-articulated problem statement and an explicit overarching question.

Thorough review of existing instruments and their contextual limitations.

Clear linkage between TANDEM’s six domains and recognised TIC frameworks.

Areas for improvement

The two sub-questions are descriptive rather than hypothesis-driven. Re-casting them as formal, directional hypotheses would sharpen the study’s focus.

Empirical evidence (e.g., pilot data) showing why existing tools under-perform in Norway would strengthen the justification for a new instrument.

Consistent terminology—decide on “course”, “programme”, or “TCP” and use it uniformly.

Cite current psychometric guidance (e.g., COSMIN responsiveness standards) to ground the work in best-practice measurement science.

Methodology: A pre-/post-survey design with 163 postgraduate students is suitable for an initial responsiveness study, and the description is detailed enough to replicate.

Strengths

The three-week course curriculum is described in detail, enabling readers to link content to TANDEM domains.

Clear reporting of recruitment procedures and response rates.

Ethical approval and informed consent are documented.

Inclusion of an exploratory factor analysis (EFA) to check dimensional stability is commendable.

Areas for improvement

The single-arm design limits causal inference. Explain why a comparison group was not feasible and temper causal language.

No attrition analysis is provided; a participant flow diagram would reveal potential bias.

Scoring excludes respondents with any missing items, which may distort change estimates. Consider multiple imputation or prorating.

Effect-size metrics (Cohen’s d, SRM) and confidence intervals are mandatory for interpreting practical significance yet are absent.

Dichotomising continuous predictors (e.g., workplace type) in regression analyses discards information and inflates error rates; retain original scales or use dummy coding.

The EFA results are densely embedded in text; a supplementary table with factor loadings and explained variance would improve clarity.

A multigroup CFA to test pre/post measurement invariance would confirm that score changes reflect true change rather than shifting factor structure.

Data-sharing plans do not yet meet journal policy—deposit de-identified data in a public or restricted-access repository.

Results: Findings are well organised and tables are easy to follow, but some critical statistical information is missing.

Strengths

Significant pre-/post gains in overall competence and five of six domains clearly support TANDEM’s sensitivity.

Regression results identifying “less prior trauma training” as the strongest predictor of learning have practical value.

Tables 3–5 concisely present descriptive and inferential statistics.

Areas for improvement

Add paired-samples Cohen’s d (or SRM) and 95 % CIs to all change estimates.

Provide baseline distribution data to substantiate the proposed ceiling effect in the ‘Readiness’ domain.

Report standardised β coefficients and CIs for regression predictors to show relative importance.

Include the full factor-loading matrix and variance explained for pre- and post-EFA, ideally in an appendix.

A simple forest plot or line graph illustrating mean change by domain would make the results more digestible.

Discussion: The discussion is thoughtful and balanced, placing findings within the wider TIC literature while candidly noting limitations.

Strengths

Plausible interpretation of the ‘Readiness’ ceiling effect and perceptual nature of gains in ‘Organisational Culture’.

Transparent acknowledgement of sample homogeneity, self-report bias, and lack of long-term follow-up.

Recommendations for further research in diverse settings are appropriate.

Areas for improvement

Address potential response-shift bias explicitly—future studies could use “then-retro” ratings or structural-equation methods.

Compare observed effect sizes (once reported) with those from ARTIC or similar instruments to situate findings quantitatively.

Temper concluding statements such as “the basis for using TANDEM is significantly strengthened” by noting design constraints.

Offer a clearer roadmap for next steps: (1) measurement-invariance testing; (2) longer-term follow-up; (3) broader professional samples; (4) objective outcome correlations.

General Comments: The manuscript is generally well written and logically organised. Minor English-usage issues (“anchoring”, “line of work”) and a few typos can be handled in copy-editing. Tables are informative but would benefit from supplementary material for psychometric detail and a visual summary of change. Aligning fully with STROBE and COSMIN guidelines—including a flow diagram and a responsiveness checklist—will raise reporting quality.

7. PLOS authors have the option to publish the peer review history of their article (what does this mean? ). If published, this will include your full peer review and any attached files.

**Do you want your identity to be public for this peer review?** For information about this choice, including consent withdrawal, please see our Privacy Policy .

Reviewer #1: No

Reviewer #3: **Yes:** Danilo Assis Pereira

---

## [Author Response · Author response to Decision Letter 3]

14 Oct 2025

Se attached "Response to reviewers (3rd revision)

---

## [Decision Letter · Decision Letter 3]

28 Oct 2025

PONE-D-24-48365R3The sensitivity of TANDEM – a new measure of trauma competencePLOS ONE

Dear Dr. Nordanger,

Thank you for submitting your manuscript to PLOS ONE. After careful consideration, we feel that it has merit but does not fully meet PLOS ONE’s publication criteria as it currently stands. Therefore, we invite you to submit a revised version of the manuscript that addresses the points raised during the review process.

Strong and relevant study; conceptually well-founded.Needs tighter consistency in sample reporting and terminology.Clarify analytic criteria and display key outputs in-paper.Justify or test sensitivity of dichotomized variables.Ensure full data-sharing, codebook, and formatting compliance.

We look forward to receiving your revised manuscript.

Kind regards,

Muhammad Shahzad Aslam, Ph.D.,M.Phil., Pharm-D

Academic Editor

PLOS ONE

Journal Requirements:

Additional Editor Comments (if provided):

The study’s positioning of TANDEM as a multidimensional measure of trauma competence is compelling. The introduction could, however, articulate more clearly how TANDEM extends existing frameworks (e.g., TICOMETER, Trauma-Informed Practice Scales). Highlighting specific psychometric or conceptual gaps filled by TANDEM would help readers grasp its unique contribution beyond incremental validity claims.Presenting the main outputs (effect sizes, confidence intervals, standardized and unstandardized coefficients) within the article proper rather than only in supplements will make the evidence base more visible and evaluable. The inclusion of data-repository links and a short codebook will further demonstrate a commitment to open science practices.

Reviewers' comments:

Reviewer's Responses to Questions

**Comments to the Author**

1. If the authors have adequately addressed your comments raised in a previous round of review and you feel that this manuscript is now acceptable for publication, you may indicate that here to bypass the “Comments to the Author” section, enter your conflict of interest statement in the “Confidential to Editor” section, and submit your "Accept" recommendation.

Reviewer #3: All comments have been addressed

2. Is the manuscript technically sound, and do the data support the conclusions?

Reviewer #3: Yes

3. Has the statistical analysis been performed appropriately and rigorously? 

Reviewer #3: Yes

4. Have the authors made all data underlying the findings in their manuscript fully available?

Reviewer #3: No

5. Is the manuscript presented in an intelligible fashion and written in standard English?

Reviewer #3: Yes

6. Review Comments to the Author

Reviewer #3: What I’m recommending to the authors: Finalize the Sample section by keeping only the updated paragraph with the 339 denominator, 167 paired responses, 4 exclusions, final N=163, and overall 48.1% response rate; remove the legacy “70%” language.

Clarify invariance testing by stating the exact decision rules (e.g., ΔCFI ≤ .01; ΔRMSEA ≤ .015; non‑significant χ² difference), and correct the “reduction in χ²” phrasing if reversed. Remove or clearly sequester the residual EFA paragraph.

Make the analysis outputs visible in‑paper. Ensure pre–post effect sizes with 95% CIs and clearly labeled standardized vs unstandardized coefficients appear in the main tables, not only in DOCX supplements. Briefly acknowledge multiplicity and whether any FDR‑style control was considered.

Justify or reduce dichotomization of Target age group, Workplace type, and Role type; if feasible, re‑run with dummy/ordered terms and report sensitivity analyses.

Harmonize terminology. Pick either “Work culture” or “Organisational/Organizational culture” and use it consistently across text and tables; choose UK or US spelling and apply uniformly.

Lock down data availability by inserting the repository DOI/URL and adding a short codebook (variables, scales, and recodes).

Align authorship metadata so the author order and corresponding author in EM match the manuscript.

Tidy presentation by removing all placeholders/duplications and ensuring the final table/figure set conforms to PLOS formatting.

7. PLOS authors have the option to publish the peer review history of their article (what does this mean? ). If published, this will include your full peer review and any attached files.

**Do you want your identity to be public for this peer review?** For information about this choice, including consent withdrawal, please see our Privacy Policy .

Reviewer #3: **Yes:** Danilo Assis Pereira

---

## [Author Response · Author response to Decision Letter 4]

3 Dec 2025

Se attached "Response to reviewers" (4rd revision)

---

## [Editor Report · Decision Letter 4]

14 Dec 2025

The sensitivity of TANDEM – a new measure of trauma competence

PONE-D-24-48365R4

Dear Dr. Nordanger,

We’re pleased to inform you that your manuscript has been judged scientifically suitable for publication and will be formally accepted for publication once it meets all outstanding technical requirements.

Kind regards,

Muhammad Shahzad Aslam, Ph.D.,M.Phil., Pharm-D

Academic Editor

PLOS One
---

## [Editor Report · Acceptance letter]

PONE-D-24-48365R4

PLOS One

Dear Dr. Nordanger,

I'm pleased to inform you that your manuscript has been deemed suitable for publication in PLOS One. Congratulations! Your manuscript is now being handed over to our production team.

Kind regards,

on behalf of

Dr. Muhammad Shahzad Aslam

Academic Editor

PLOS One